# Kinetics of Cellular Cobalamin Uptake and Conversion: Comparison of Aquo/Hydroxocobalamin to Cyanocobalamin

**DOI:** 10.3390/nu16030378

**Published:** 2024-01-27

**Authors:** Sergey N. Fedosov, Ebba Nexo, Christian W. Heegaard

**Affiliations:** 1Department of Molecular Biology and Genetics, Aarhus University, 8000 Aarhus C, Denmark; cwh@mbg.au.dk; 2Department of Clinical Medicine/Clinical Biochemistry, Aarhus University Hospital, 8200 Aarhus N, Denmark; enexo@clin.au.dk

**Keywords:** vitamin B_12_, cobalamin, kinetics, uptake, HeLa cells

## Abstract

Cyanocobalamin (CNCbl) and aquo/hydroxocobalamin (HOCbl) are the forms of vitamin B_12_ that are most commonly used for supplementation. They are both converted to methylcobalamin (MeCbl) and 5′-deoxyadenosylcobalamin (AdoCbl), which metabolize homocysteine and methylmalonic acid, respectively. Here, we compare the kinetics of uptake and the intracellular transformations of radiolabeled CNCbl vs. HOCbl in HeLa cells. More HOCbl was accumulated over 4–48 h, but further extrapolation indicated similar uptake (>90%) for both vitamin forms. The initially synthesized coenzyme was MeCbl, which noticeably exceeded AdoCbl during 48 h. Yet, the synthesis of AdoCbl accelerated, and the predicted final levels of Cbls were MeCbl ≈ AdoCbl ≈ 40% and HOCbl ≈ 20%. The designed kinetic model revealed the same patterns of the uptake and turnover for CNCbl and HOCbl, apart from two steps. First, the “activating” intracellular processing of the internalized HOCbl was six-fold faster. Second, the detachment rates from the cell surface (when the “excessive” Cbl-molecules were refluxed into the external medium) related as 4:1 for CNCbl vs. HOCbl. This gave a two-fold faster cellular accumulation and processing of HOCbl vs. CNCbl. In medical terms, our data suggest (i) an earlier response to the treatment of Cbl-deficiency with HOCbl, and (ii) the manifestation of a successful treatment initially as a decrease in homocysteine.

## 1. Introduction

Cobalamin (Cbl, B_12_) is an important nutrient required at a quantity of 2–3 µg per day [1,2,3]. It is synthesized by certain bacteria and archaea present in open environments [4] and the digestive systems of herbivores [5]. The ingested or internally produced Cbl enters the circulation with the help of several transporting proteins [6,7], which mediate Cbl uptake from the intestine. The ingestion of a normal physiological dose (2–10 µg) results in the maximal absorption of 1.5–2 µg via the specific transportation route [2,6,7]. The high oral doses of 1–5 mg are used for the treatment of Cbl deficiency and add a further 10–50 µg to the specific uptake, which penetrate the intestinal walls via diffusion [2,3,6]. The final step of Cbl delivery goes via the blood stream, where a specific protein-transporter transcobalamin (TC) binds Cbl. Afterward, the TC–Cbl complex (holoTC) is internalized by the tissues with the help of the ubiquitously expressed cell surface receptor CD320 [6,7,8].

Cbl exists in several forms, differing from each other by the exchangeable group X, coordinated to the cobalt site of the vitamin X··[Co^3+^]Cbl [3,9,10]. The most common variants of Cbl include cyanocobalamin (CNCbl), aquo/hydroxocobalamin (HOCbl), methylcobalamin (MeCbl), and 5′-deoxyadenosylcobalamin (AdoCbl). HOCbl, MeCbl, and AdoCbl naturally occur in most animal products, where only trace quantities of CNCbl are usually found [1]. The latter form is a synthetic compound commonly used for food fortification and supplementation due to its chemical stability and low cost. The affinity of X ligands for the cobalt site varies. Thus, Me, Ado, and CN groups belong to the strongest ligands, while the water in HOCbl (at pH < 7) is the weakest [9,10].

Irrespective of their original X groups, all XCbls are internalized and similarly processed by animal cells [11,12]. This route is roughly outlined in Figure 1, where we generally adopt the nomenclature of Banerjee et al. [12]. At the first step, the enzyme chaperon CblC [12] (encoded by gene MMACH) reduces the incoming X[Co^3+^]Cbl to [Co^2+^]Cbl. The reduction of X[Co^3+^]Cbl requires the help of methionine synthase reductase (MSR/CblE) and NADPH, or glutathione (GSH), or both [13]. When cobalt ion gains additional electron(s), XCbl looses its original X-group [12], with the exception of a few artificial nonconvertible anti-vitamins, which are incapable of forming coenzymes [14]. The unprocessed XCbl (either excessively absorbed or resistant to CblC) is cleared from the cell by the multidrug resistance protein 1 transporter (MRP1), encoded by the ABCC1 gene [15,16], and notated as ABC in Figure 1.

The intracellular activation of the generated [Co^2+^]Cbl continues with the help of the adapter protein CblD [12,17]. The latter transports Cbl to either (i) cytoplasmic methionine synthase (MS), assisted by MS reductase (CblE); or (ii) methylmalonyl-CoA mutase (MMCM), assisted by adenosyltrasferase (CblB). The first transportation route ends by the synthesis of the coenzyme MeCbl, which mediates the transfer of the methyl from methyltetrahydrofolate to homocysteine [12,18]. The second route leads to the mitochondria, where the formed AdoCbl serves as a coenzyme of MMCM and helps to convert a nonmetabolizable product of fatty/amino acid degradation methylmalonyl-CoA to succinyl-CoA [12,18]. Both enzymatic reactions are critical for human health [2,3]. Thus, a decreased activity of MS results in a low level of the major methylating precursor methionine, as well as in a high level of unmetabolized homocysteine [1,2,3]. The latter thiol is frequently associated with inflammation, oxidative stress, and microvascular diseases [2]. In addition, the trapping of folate in its “inactive” methyltetrahydrofolate form hampers DNA synthesis [2,3]. Low activity of the second Cbl-dependent enzyme MMCM leads to the accumulation of methylmalonic acid and a general acidification of the body [2,3].

Unclarified issues regarding Cbl uptake and processing remain. For example, the overall retention of Cbl in the cells seemingly depends on its original X group. Thus, a two-fold increase in the accumulation of HOCbl and MeCbl compared to CNCbl has been reported for several cell types [19,20,21]. This difference was not associated with a better recognition of the TC–HOCbl complex by its receptor on the cell surface [20], and the underlying mechanisms remain unknown. It seems that the efficiency of retention is associated with the reductive activity of CblC, because the CblC-mutated cells showed a decreased uptake/retention of CNCbl and alkylcobalamins in particular [11,19], while the accumulation of HOCbl was hampered to a much lower degree [19]. Another example concerns a nonreducible “antivitamin” ethylphenyl-Cbl, which remained in the extracellular circulation and showed a low tissue accumulation in mice [22].

A puzzling dispersion in the proportions between MeCbl and AdoCbl was reported for cell cultures [11,19,20,21,23,24], while animal and human tissues seemingly exposed a great prevalence of endogenous AdoCbl above MeCbl [25,26,27]. Noteworthy, we have recently demonstrated that several widely used methods for the tissue extraction of Cbl can convert MeCbl to HOCbl, thereby masking the true balance of intracellular Cbl coenzymes [28].

Uncertainties about the efficacy of the uptake of XCbl and the order of coenzyme appearance complicate the choice of the XCbl form to be used for the treatment of Cbl deficiency. Here, we scrutinize the cellular uptake and conversion of Cbl by (i) monitoring the accumulation of radioactivity from either CN [^57^Co]Cbl or HO[^57^Co]Cbl by HeLa cells; and (ii) analyzing the kinetics of the transformation of the original ligands to the Cbl coenzymes according to a designed computer model. We found a faster processing of HOCbl in comparison to CNCbl, and we also found that MeCbl appeared as the first intracellular coenzyme, irrespective of the supplied vitamin form.

## 2. Materials and Methods

### 2.1. Materials

All salts and standard reagents were purchased from Sigma-Aldrich (St. Louis, MO, USA). The radioactive tracer CN[^57^Co]Cbl was obtained from MP Biomedicals LLS (Solon, OH, USA). It was converted to HO[^57^Co]Cbl by photo-aquation [1,9], according to the procedure described elsewhere [29]. The isotope ^57^Co emits γ-radiation (t_½_ = 272 days), which was counted in the scintillation cocktail OptiPhase HiSafe 3 (Perkin Elmer, Hopkinton, MA, USA) using the energy window of 0–200 keV. The apo-form of bovine transcobalamin (bTC) originated from a store produced earlier, [7] and ref. thereof. HeLa cells (CCL-2) were purchased from the American Type Culture Collection. Tissue culture materials included Dulbecco’s modified Eagle’s medium (DMEM) without B_12_; fetal bovine serum, trypsin/EDTA solution, Penicillin–Streptomycin mixture, and Dulbecco’s phosphate-buffered saline (DPBS) were all bought from Merck Life Science A/S, Søborg, Denmark.

### 2.2. Accumulation of Radioactive ^57^Cbl in HeLa Cells

The HeLa were cultured at 37 °C in a 5% CO_2_ atmosphere using a Dulbecco’s Modified Eagle’s medium (DMEM) (Gibco™) supplemented with 10% heat-inactivated fetal bovine serum (Sigma-Aldrich), final B_12_ ≈ 15 pM, free binding capacity ≈ 20 pM. The cells were incubated in the medium containing either CN[^57^Co]Cbl or HO[^57^Co]Cbl (final concentration of 0.22 nM, both pre-bound to bTC = 0.44 nM). A repeated addition of the unsaturated bTC alone (0.44 nM) was conducted after 24 h. The incubation was continued for 4, 12, 24, and 48 h, whereupon the media were collected for further analysis. The cells on well-plates were gently rinsed twice with Dulbecco’s phosphate-buffered saline (DPBS) and detached with the trypsin/EDTA solution in DPBS without CaCl_2_ and MgCl_2_ (Gibco™). The completion of the cell detachment was verified by microscopy. The cell suspensions were centrifuged at 4 °C in a microfuge for 5 min at 100× *g* to pellet the cells. The supernatants (containing the released surface Cbl) were collected, and the cell precipitates (containing the intracellular Cbl) were resuspended in a minimal volume of DPBS (0.1–0.2 mL). Both the supernatants and the cell suspensions were used for the further analysis of radioactivity, interpreting them as the fractions of the cell surface and intracellular ^57^Cbl. The number of cells in each sample was counted in the automated cell counter Moxi Z (Orflo, Plainfield, IN, USA) using a suitable dilution of the original suspension.

### 2.3. Extraction of Cbls from Cells

A phenol–chloroform mixture (volumes 1:1) was used for the extraction of Cbl from the cells [1], as well as for the denaturing of proteins and the dissolving of membranes. Prior to the application of the organic mixture, it was vigorously shaken with an equal volume of water; the emulsion was settled, whereupon the main part of the upper aqueous layer was discarded. This procedure was repeated 4–5 times in order to decrease the acidity of the organic phase and saturate it with water. The ready mixture was stored at 4 °C.

All extraction procedures were carried out in a dim light to prevent the conversion of MeCbl and AdoCbl to HOCbl. The original cell suspension of ≈0.15 mL (or a model sample) was placed in a 2 mL vial whereupon 0.8 mL of 0.32 M NH_4_Acetate buffer, pH 4.6, was added. The mixture was vigorously shaken with 0.5 mL of phenol–chloroform (5–6 times over ≈2 min) and centrifuged for 3 min (20,000× *g*, room temperature) to form an explicit layer of denatured proteins between the bottom organic phase and the top aqueous phase. Less than 0.5% of the total radioactivity was removed with the discarded aqueous solution. The phenol–chloroform phase was washed with 1.4 mL of water by an analogous procedure (time of centrifugation was decreased to 1 min). The loss of radioactivity was ≈0.7%. The organic phase (0.5 mL) was mixed with equal volumes of chloroform and acetone, followed by the addition of 0.4 mL 0.32 M NH_4_Acetate buffer, pH 4.6, containing 1 mM 5-aminotetrazole (ATZ). The sample was vigorously shaken and incubated for 20–24 h at 37 °C under rotation and complete protection from light. The presence of a “reversible” ligand ATZ with a reasonably high affinity for the Co-site ([7] and refs. Thereof) decreased by >75% association of HOCbl with the middle layer of denatured proteins during the backward transition of Cbl from the organic to aqueous phase in a model experiment. The effect was caused by a gradual dissociation of unspecific protein–Cbl coordination complexes, e.g., protein–His··[Co^3+^]Cbl [7,9], and the formation of soluble ATZ··Cbl. The sample was centrifuged (1 min, 20,000× g), the upper phase was collected, and the lower phase was washed with 0.2 mL of water. The wash water was pooled (after centrifugation) with the main aqueous fraction. The loss of radioactivity (still associated with the well-separated middle layer of denatured proteins) corresponded to ≈4.5%.

The collected aqueous phase (containing ≈ 95% of radioactivity) was vigorously shaken several times with 1 mL of diethyl ether to extract the remaining phenol, whereupon the upper organic phase was discarded with no accountable loss of radioactivity. The main part of diethyl ether in the bottom aqueous solution was removed by blowing a hairdryer over the samples (3 × 1 min with 1 min intervals to prevent bubbling). The sample was lyophilized without heating in a centrifugal evaporator (1 day in total darkness), whereupon the dry solids were dissolved in 0.15–0.25 mL of 0.2 M NH_4_Acetate buffer, pH 9.3, and heated for 10 min at 95 °C to convert the potentially present HOCbl, ATZCbl, and GSCbl (with reversibly bound Co-ligands) to a more stable NH_3_Cbl [9]. The formation of the latter complex improved the separation of the original HO/ATZCbl forms from CNCbl on HPLC, see Section 2.5. An individual test with pure Cbls showed that the ammonium treatment converted most of the GSCbl and all HOCbl to NH_3_Cbl, but this procedure did not affect the fractions of CNCbl, AdoCbl, MeCbl, and SO_3_Cbl, which contain high-affinity ligands bound to the Co-site. The ready sample was centrifuged (10 min, 20,000× *g*) and the supernatant was collected.

### 2.4. Measurements of Radioactivity

The isotope ^57^Co emits γ-radiation (t_½_ = 272 days), which was counted in the scintillation cocktail OptiPhase HiSafe 3 using the energy window of 0–200 keV. For this purpose, 1 mL of a solution containing ^57^Cbl (in the cell medium/0.15 M NaCl/0.3 M NH_4_Acetate buffer pH 4.6/phenol-chloroform/fractions from HPLC) was mixed with 4 mL of the scintillation cocktail, whereupon the mixture was counted for 2 min. Most solutions did not interfere with the scintillation counting, except for the cell medium with phenol red added (which decreased the signal by factor ×0.72), the phenol–chloroform mixture (×0.60), and the phenol–chloroform–acetone mixture (×0.50). These “quenching factors” were taken into account. A typical background count corresponded to 20 CPM, while the span of recorded signals usually covered 300–15,000 CPM. The counting of the experimental samples required 1/5 of the collected cell medium; 1/10 of the supernatant after trypsin treatment; and 1/10 of the cell suspension (if we enumerate the most important objects of our study).

### 2.5. HPLC Separation of Cbls

The separation of Cbls by HPLC was performed on a reverse phase column Hypersil C-18, 250 × 4 mm, 120 Å, 5 µm, 170 m^2^/g (Thermofisher) pre-warmed to 40 °C, flow 1 mL/min. A mixture of pure Cbls (10 μM) or the final cell extracts (all 120–160 μL) were injected into an HPLC system. The separating gradient included two solvents: A (0.2 M NH_4_Acetate buffer, pH 4.6) and B (A + 80% ethanol). They were mixed at the indicated time intervals as follows: (0 min, 12% B), (30 min, 30% B), (33 min, 100% B), (45 min, 12% B), (55 min, 12% B), with linear transitions between these points. The commercial Cbl samples (10-20 μM) were detected on the elution profile by their optical density (254 nm). They showed the following retention times: GSCbl (7 min), SO_3_Cbl (8.5 min), NH_3_Cbl (10 min), ATZ and HOCbl (11.5 min), CNCbl (13 min), AdoCbl (18.5 min), MeCbl (26 min). The Cbls with non-conventional groups (GS/SO_3_/NH_3_/ATZ) were prepared by incubating HOCbl with some excess of the particular cobalt-coordinating ligand (e.g., 10 μM HOCbl + 20 μM GSH). The content of different radioactive Cbls in cell extracts was determined by collecting 1 mL fractions during HPLC separation and measuring their radioactivity as described above.

### 2.6. Computer Modeling

The design of the model is thoroughly discussed in Section 3.5, Section 4.2, Section 4.3, Section 4.4, Section 4.5, Section 4.6, and Appendix A. The freeware Biochemical System Simulator COPASI version 4.3.6 (http://copasi.org/, accessed on 21 December 2023) was used for computer simulations, see Section 3.5. The concentrations of reactants (all in percent units) at the starting point (incubation time 0 h) were as follows: [R_out_] = 40%, corresponding to the apparent concentration of holoTC receptor on the cell surface; [C_out_] = 100%, the extracellular radioactive cobalamin X[^57^Co^3+^]Cbl bound to TC (i.e., TC-X[^57^Co^3+^]Cbl complex); [C_in_] = 0%, the unprocessed radioactive X[^57^Co^3+^]Cbl inside the cells; [C2] = 0%, the processed radioactive [^57^Co^2+^]Cbl inside the cells; [C_surf_] = 0%, the unprocessed radioactive X[^57^Co^3+^]Cbl recirculated from the intracellular space to the cell surface; [MC] = 0%, the freshly synthesized MeCbl; [AC] = 0%, the freshly synthesized AdoCbl. The differential equations, which can be used for simulations with the help of any suitable software, are given in Appendix A. Stepwise adjustments of the initially chosen rate coefficients were initially completed by a visual comparison of the simulated curves with the experimental points. The final adjustment of coefficients, as well as the verification of the model, involved the analysis of relative residuals (rr = r/σ), i.e., residuals r = (y_i_ − f_(xi)_) related to the assessed experimental dispersion σ. Adjustments were continued until no systematic deviation of rr-values from 0 at any time was detected, see Appendix A for further technical details.

## 3. Results

### 3.1. Exploratory Tests of Cbl Uptake by HeLa Cells (24 h)

HeLa cells with similar seeding densities were incubated in the culture media containing either radioactive CN[^57^Co]Cbl or HO[^57^Co]Cbl (0.22 nM), both pre-bound to a surplus of bovine TC (bTC = 0.44 nM). At the end of incubation (24 h), no visual difference in the cell number was observed for the two setups (though the cells were not counted). The pilot assay showed a significant difference between the intracellular radioactivity when supplying the cells with CN[^57^Co]Cbl (≈29%) vs. HO[^57^Co]Cbl (≈58%), see Table 1. The same was the case for the residual radioactivity in the respective cell-free mediums (≈47% vs. ≈12%). The representation of both ligands on the cell surface (spanning from 5% to 12%) was similar. The unaccounted loss of radioactivity (12–26% in both cases) was most probably caused by the reversible (?) adsorption of the bTC–^57^Cbl complex on the walls of the incubation chamber. The tendency of bTC–^57^Cbl to bind to the cell culture plates was very pronounced (≈80% adsorption) if no excessive protein was present in the incubation medium. However, the addition of bovine serum (final concentration 10%) significantly counteracted bTC–^57^Cbl surface adsorption. The latter setup was used in all experiments presented in this work, including the pilot assay (Table 1).

### 3.2. Time-Dependent Accumulation of Cbl in HeLa Cells

The cellular growth and accumulation of radioactivity in the presence of either CN[^57^Co]Cbl or HO[^57^Co]Cbl (both bound to a surplus of bTC) was examined at several time intervals. More unsaturated bTC was added to the medium at 24 h to compensate its loss, after the unprocessed Cbl had been excreted to the medium and repeatedly delivered to the cells with help of bTC (see Section 3.5).

The number of seeded cells increased from n = 0.8 × 10^7^ (at 4 h) to 1.0 × 10^7^ (at 24 h), but then decreased to 0.9 × 10^7^ (48 h) due to the cell detachment from the cell culture plates. This tendency accelerated over time, and after 3 and 4 days of incubation the residual cell numbers were decreased to below 0.5 × 10^7^ (see also Discussion, Section 4.1). Therefore, we limited our assay to the time span of 4–48 h and normalized both the intracellular and the surface-bound radioactivity to a “standard” number of cells (n = 1 × 10^7^, found at 24 h) to compare the samples collected at different time points. This means that the radioactive counts of the intracellular and surface fractions were divided by factors 0.8, 0.9, and 1.0 for the values of n = 0.8 × 10^7^, 0.9 × 10^7^, and 1.0 × 10^7^, respectively. The counts are given in percent, where 100% corresponds to the total radioactivity added to the medium at the start.

The normalized records of intracellular radioactivity were plotted over time, and the results are shown in Figure 2a. The dataset with counted cells was supplemented by the pilot study from Table 1, conducted identically but without the actual cell count. These points are indicated as closed symbols in Figure 2a. Each point shows the result for a particular cell well. The average dispersion (σ) of the experimental points was assessed from the repeated measurements at 24 h and 48 h, see the panel legend and Appendix A. A noticeable difference between the datasets for CNCbl and HOCbl was found on the interval of 12–48 h. Thus, the ratios of intracellular radioactivity (HO[^57^Co]Cbl/CN[^57^Co]Cbl) were significantly above 1.0 according to a *t*-test: mean ± SEM = 1.88 ± 0.08 (*p* = 3 × 10^−5^). This observation required the introduction of different processing rate constants (k_6_) for each ligand in the overall kinetic model, see Section 3.5 and Section 4.5.

The time records of the tightly bound surface Cbl (removed only after trypsinolysis of the cells) are presented in Figure 2b (including the exploratory tests from Table 1). The average dispersion of points in this panel was assessed as explained in Appendix A. The final kinetic model (solid lines) gave a reasonable approximation of the experimental data. The identity of real and theoretical points was assessed as *p* = 0.66 for the pooled CNCbl and HOCbl datasets in Figure 2, see Appendix A. The necessity of different surface detachment rate constants (k_5_) for CNCbl and HOCbl is illustrated by a significantly deviating dashed curve, where equal detachment rates for both ligands were assumed (k_5HO_ = k_5CN_). In this example, the identity of real and theoretical points was very low (*p* = 3 × 10^−4^ according to *t*-test of residuals). Figure 2b also shows the theoretical simulations for the turnover of TC-Cbl receptor (R) present on the cell surface (dotted curves). Further details are clarified in Section 3.5 and Section 4.4.

### 3.3. Intracellular Cbls and The Extraction and Analysis of Cbl Forms by HPLC

We used a modification of phenol–chloroform extraction method. This procedure preserved the original composition of Cbls in an artificial mixture, containing GSCbl, CNCbl, HOCbl, AdoCbl, and MeCbl in blood plasma with 10 mM glutathione (GSH), see Appendix A. GSH is capable of converting MeCbl, CNCbl, and HOCbl to GSCbl, if all heated together [28,30]. Our tests showed that GSH did not enter the phenol–chloroform phase and could not disturb the original balance of Cbls.

The observed extraction yields were quite high: mean ± SEM = 91.5 ± 1.4% for CN[^57^Co]Cbl setup (n = 8) and 92.3 ± 1.0% for HO[^57^Co]Cbl setup (n = 8). The cell extracts after incubation with either CN[^57^Co]Cbl or HO[^57^Co]Cbl (corresponding to the experimental points in Figure 2a) were analyzed by HPLC, see Methods, Section 2.5, and ref. [31].

The profiles of Cbls in cell extracts are presented in Figure 3. All records were normalized, i.e., the radioactivity of each HPLC fraction is related to the total radioactivity in the profile. The mean record of the two 48 h profiles (obtained in two different cell experiments) is shown for both CNCbl (Figure 3a) and HOCbl (Figure 3b) setups. The material balance of radioactivity in the collected fraction vs. the injected sample corresponded to mean ± SEM = 81 ± 1% for CN[^57^Co]Cbl (n = 5) and 85 ± 3% for HO[^57^Co]Cbl (n = 5), probably reflecting a small loss in the injection loop. The well-identifiable Cbl peaks contained 80 ± 2% of all the radioactivity in the collected fractions.

The profiles of the CN[^57^Co]Cbl samples are depicted in Figure 3a and show the dominance of the original vitamin at 4 h of incubation. It gradually disappeared in the cells harvested after 12, 24, and 48 h of incubation, when the accumulation of HO[^57^Co]Cbl (corresponding to NH_3_[^57^Co]Cbl in Figure 3a) and MeCbl was clearly detected. In contrast, the increase in AdoCbl was much slower.

A similar picture emerged for the intracellular conversion of HO[^57^Co]Cbl (Figure 3b), irrespective of its better accumulation in the cells. Some ambiguity was associated with the initial part of the 4 h record (Figure 3b), where a noticeable breakthrough peak of radioactivity was recorded at 1.5 min elution. This abnormal peak (12%) exceeded the usual breakthrough fractions of 3–5%, and it was absent in the analogous 4 h sample for CN[^57^Co]Cbl. We attributed the excessive size of this peak to a strong coordination of HO[^57^Co]Cbl via its cobalt site to a minor fraction of hydrophilic compounds, resistant to ammonium treatment. Such compounds can be peptides with thiol groups or more potent sulfate groups. When the cellular uptake of HO[^57^Co]Cbl increased after 12, 24, and 48 h of incubation, the percentage of the ambiguous fraction fell to the usual 3–5% (Figure 3b), implying its low representation. Based on the aforementioned reasons, the excessive part of this breakthrough peak (4 h in Figure 3b was assigned to the NH_3_[^57^Co]Cbl fraction during the kinetic analysis.

### 3.4. Transformations of the Intracellular Cbls over Time

The curves of the intracellular processing of Cbl (Figure 4) were reconstructed using the peak fractions of the HPLC records (Figure 3) applied to the total intracellular radioactivity at the respective incubation periods (Figure 2a). The obtained results are presented as percentages, which represent a fraction of the total ^57^Cbl (added to the medium) transformed to each particular Cbl inside the cells. The standard deviation of the experimental measurements is given in the figure legend (see also Appendix A). The percent units can be recalculated to the actual intracellular concentration of each individual Cbl form. For instance, an intracellular XCbl assessed as 10% corresponds to 0.11 pmol of the originally added ^57^Cbl (1.1 pmol). This value should be divided by the total volume of HeLa cells, assessed as 25 × 10^−6^ L (according to a single cell volume ≈ 2500 μm^3^ [31] multiplied by the total number of cells, n = 10^7^). In this way, 10% of this hypothetical XCbl corresponds to 4.4 nM of its intracellular concentration. Further explanations follow in Section 3.5.

We tested the “leakage” of the processed Cbls from cells at 24 h of incubation with CN[^57^Co]Cbl by analyzing Cbl forms in the medium. Only CN[^57^Co]Cbl was found, with no trace of HO[^57^Co]Cbl, Me[^57^Co]Cbl or Ado[^57^Co]Cbl present.

### 3.5. Kinetic Model of Cbl Uptake and Its Intracellular Transformations

The experimental data were used to design a kinetic model, adequately approximating the time curves of (i) the accumulated intracellular Cbl (Figure 2a), (ii) the surface-bound Cbl (Figure 2b), and (iii) the intracellular conversions of CN[^57^Co]Cbl (Figure 4a) and HO[^57^Co]Cbl (Figure 4b).

The model includes eight steps (Figure 5), and it is a simplification of a more complex process. The model uses relative units, where 100% corresponds to 0.22 nM concentration of ^57^Cbl in the external medium of 5 mL (i.e., quantity of 1.1 pmol), bound to bTC = 0.44 nM (replenished by adding more bTC = 0.44 nM at 24 h). All units for other interacting compounds (e.g., R) are related to the concentration of ^57^Cbl in the medium, and, e.g., R = 20% corresponds to the R-concentration of 0.044 nM. The concentrations of reactants, rates, and rate constants refer to the whole reaction compartment (5 mL) and do not scale these values upon the transition of Cbl to and from the cells (10^7^ HeLa cells with a volume of ≈25 µL). Therefore, a direct comparison of the obtained kinetic parameters is possible for either extracellular or intracellular compartment, or when aligning the identical steps for CN[^57^Co]Cbl vs. HO[^57^Co]Cbl.

**Figure 5 nutrients-16-00378-f005:**
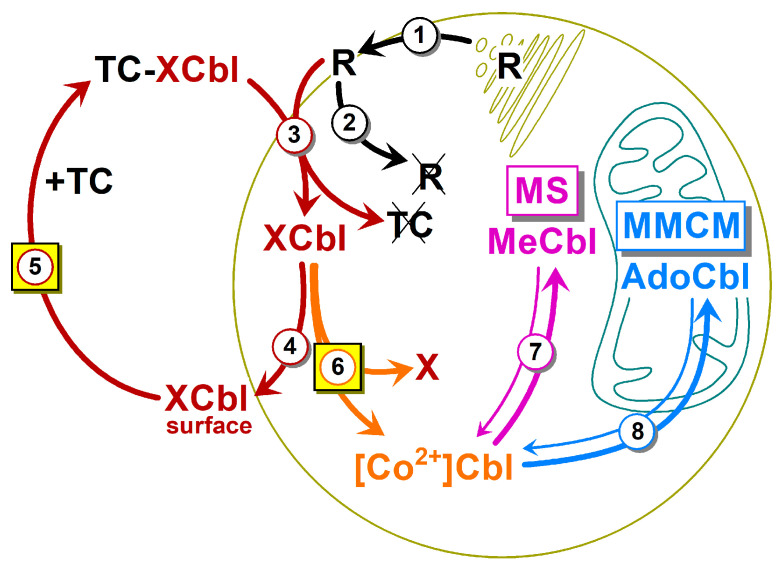
The kinetic scheme of the final model. The following steps were considered: (1) the export of receptor R (specific for TC-XCbl) to the cell surface; (2) the clearance of “empty” R from the surface; (3) the R-mediated uptake of TC-XCbl followed by the degradation of the protein moieties and the appearance of XCbl in the cytoplasm; (4) the recirculation of the unprocessed XCbl to the cell surface; (5) the dissociation of XCbl from the cells and its binding to surplus TC; (6) the intracellular reduction of absorbed XCbl to [Co^2+^]Cbl; (7) the synthesis of MeCbl in the cytoplasm; (8) the synthesis of AdoCbl in the mitochondria. Steps 7 and 8 describe the equilibriums between MeCbl/AdoCbl and [Co^2+^]Cbl. All parameters (Table 2) for CNCbl and HOCbl schemes are identical except for steps 5 and 6, as highlighted by yellow squares. See the main text for further details.

The model approximated 81 measured points (excluding zero points) shown in Figure 2 and Figure 4. The fitting utilized 10 + 2 parameter values, covering ten parameters of the CNCbl-fit plus two changed parameters of the HOCbl-fit. The other eight coefficients applied to the HOCbl dataset were reused according to the CNCbl approximation. This roughly corresponds to the assessment of each parameter value on the basis of seven points.

**Table 2 nutrients-16-00378-t002:** Parameters of the kinetic model in Figure 5.

Step	Parameter	Value	Step	Parameter	Value
1	v_1_ (R synthesis)	4% h^−1^	*6_CN_*	*k*_6_ *(reduction to Co^2+^)*	*0.27* h^−1^
2	k_2_ (futile R-clearance)	0.1 h^−1^	*6_OH_*	*k* _6_ *(reduction to Co^2+^)*	*1.8* h^−1^
3	k_3_ (R+TC-Cbl uptake)	0.003% h^−1^	7+	k_+7_ (MeCbl synthesis)	0.2 h^−1^
4	k_4_ (Cbl cell → surface)	0.5 h^−1^	7−	k_−7_ (MeCbl→ [Co^2+^])	0.1 h^−1^
*5_CN_*	*k_5_ (Cbl→ medium + TC)*	*0.16* h^−1^	8+	k_+8_ (AdoCbl synthesis)	0.02 h^−1^
*5_OH_*	*k_5_ (Cbl→ medium + TC)*	*0.04* h^−1^	8−	k_−8_ (AdoCbl→ [Co^2+^])	0.01 h^−1^

100% corresponds to 0.22 nM (or 44 nM) concentration of a compound in the incubation medium (or the intracellular compartment). Steps 5 and 6 are italicized to highlight that they have different kinetic parameters in the processing/turnover of CNCbl and HOCbl.

The optimal set of coefficients (Table 2) was reached by trial and error. It should be noted that different parameters have a different impact on the approximation of the data in Figure 2 and Figure 3. Thus, the rate constants of steps 7 and 8 are of no consequence for the cellular uptake of Cbl (Figure 2a) and its surface representation (Figure 2b). A futile turnover of R (steps 1 and 2) effects only the initial steady state concentration of R on cell surface and determines the initial jump in the uptake curves (the first phase in Figure 2a). The proportion between steps 4 and 6 has a large influence on the overall retention of Cbl, thereby affecting the second phase in Figure 3a. However, all the above steps are not important for the intracellular ratios of MeCbl, AdoCbl, and [Co^2+^]Cbl, determined exclusively by steps 7 and 8. After the initial visual approximation of the points in Figure 2a,b and Figure 4a,b, the model was finally adjusted and validated using the analysis of residuals, as described in detail in Appendix A. The final model revealed a high probability of its overlap with the experimental points (*p* = 0.79) considering all the data. A detailed survey of the model is presented in Section 4.2, Section 4.3, Section 4.4, Section 4.5 and Section 4.6.

## 4. Discussion

### 4.1. General Observations

We analyzed the cellular uptake of radiolabeled CN[^57^Co]Cbl or HO[^57^Co]Cbl by HeLa cells (Figure 2) and monitored the conversions of these vitamin forms to the newly synthesized coenzymes Me[^57^Co]Cbl and Ado[^57^Co]Cbl (Figure 4). We confirm a faster accumulation of HOCbl in comparison to CNCbl [19,20,21], at least during 2 days of incubation (Figure 2). However, our extrapolation of the time-dependent curves points to similar final levels for both forms of vitamin B_12_, apparently reached after 4–7 days. We did not proceed our experimental study beyond 48 h, because afterward, the cells started to detach from their support. The spontaneous detachment of adherent HeLa cells grown on plates is frequently observed (especially at a high cell density) and is related to their confluence, proliferation, and propensity to form metastases [32]. As these processes hinder the accurate material balance of Cbl uptake, our analysis was confined to 4–48 h.

A mild extraction procedure exposed the prevalence of Me[^57^Co]Cbl over Ado[^57^Co]Cbl during 48 h of incubation, irrespective of the starting form of the vitamin (either CN[^57^Co]Cbl or HO[^57^Co]Cbl). This observation is concurrent with earlier studies [19,23], but contradicts others [11,20], where a low representation of Me[^57^Co]Cbl was found. Some publications reported both high and low ratios of AdoCbl/MeCbl, observed in different control cell lines [21,24]. The possible reasons for such conspicuous discrepancies were thoroughly discussed earlier [11] and cover the incubation time of cultured cells, cell-type specificity, and the composition of the culture medium. Here, we would like to attract attention to a recent observation that the conditions of extraction can critically affect the balance between AdoCbl and MeCbl due to the conversion of MeCbl → HOCbl, which takes place during a harsh heating in the presence of endogenous reductants (e.g., thiols like GSH) [28]. Importantly, “a cold extraction” should not be accompanied by “a hot evaporation” of the liquid phase, because 16 h at 45 °C would match approximately 1 h at 85 °C. Therefore, no heating is advised during lyophilization in a centrifugal evaporator.

The reasons for a better cellular uptake of HOCbl in comparison to CNCbl has remained unresolved so far. Therefore, we designed a simplified kinetic model to expose the quantitative differences in the intracellular turnovers of CNCbl and HOCbl. The scarcity of the data on this subject is surprising. Thus, the current study is the second presentation of kinetic curves for the transformation of CNCbl to other Cbl forms (ref. [19] being the first work) and the first data for the conversions of HOCbl. Moreover, this is the first attempt ever performed to quantitatively interpret the curves of Cbl transformations in vivo. We had no intention of building a comprehensive model and covering all the details of the Cbl turnover. Our aim has been to create a plausible network of reactions, which may serve as a framework to explore the impact of a particular step, related to the initial Cbl processing and apparently associated with CblC in combination other assisting enzymes and cofactors.

### 4.2. Kinetic Model: Fusion of Several Metabolic Steps into One

We can expect that nearly each step of the scheme in Figure 5 incorporates several stages. Such an approach does not represent anything unusual and is frequently used in metabolic modeling, see, e.g., [33,34]. The reasons of such a fusion or “lumping” [33] of multiple steps in connection to the current study are illustrated below, using the uptake of extracellular TC–Cbl as a convenient example, see Step 3 in Figure 5.

The internalization of the TC–Cbl complex by a cell is undoubtedly a multi-step process with at least three major phases: (i) the interaction between TC–Cbl and its receptor R, followed by their internalization into an endosome; (ii) the fusion of the TC–Cbl-containing endosome with a lysosome; and (iii) the destruction of TC and R, followed by the release of Cbl to the cytoplasm [6,7,8]. In all these transitions, Cbl exists in its original vitamin form (e.g., CNCbl), which precludes the differentiation between the aforementioned steps in the frame of our study, except for the discrimination between the extracellular and intracellular vitamin. Therefore, all fractions of the freshly absorbed Cbl (spread across different intracellular compartments) were pooled and regarded as a single intermediate. We see no problem in such a presentation. It follows strict kinetic rules and provides a reasonably close approximation of the extended models, as is proven in Appendix A.

In the next sections, we attempt to translate all kinetic constants into biochemical and physiological terms.

### 4.3. Kinetic Model: Steps 1–3, the Circulation of the Surface Receptor R, and the Internalization of Cbl

The assumed rate (v_1_) of receptor R synthesis and the constant (k_2_) of its futile clearance in the absence of TC–Cbl provide the initial surface level of R = 40%, see Figure 2b, starting point. This value can be interpreted as the apparent concentration of R = 88 pM (see Section 3.5 concerning the connection between % and the real concentrations or quantities). This R-concentration corresponds to a layer of HeLa cells (n = 10^7^) exposed to the medium of 5 mL. The clearance of TC–Cbl complex decreased R to 15% ≈ 33 pM at 10 h of incubation according to the model simulations in Figure 2b. The equal potency of TC–CNCbl and TC–HOCbl complexes was assumed during their interactions with R, as follows from the literature [20]. Later on, a gradual restoration of the original R-level followed (Figure 2b). It should be noted that the curves of R are not identical for CNCbl and HOCbl in the time range of 10–50 h (Figure 2b). This difference is caused by a higher concentration of TC–CNCbl in the medium, maintained at an elevated level because the intracellular processing of CNCbl is not as efficient as for HOCbl (step 6 in Figure 5), and a considerable part of the accumulating CNCbl is excreted (step 4 in Figure 5). The recirculated CNCbl binds to TC, whereupon TC–CNCbl (C_out_) drives R_out_ into the cell, according to the law of mass action for the reaction C_out_ + R_out_ → C_in_ + R_in_.

We did not consider the formation of a complex R··TC–Cbl, undoubtedly present, but not significantly adding to the accumulation of Cbl on the cell surface, see steps 4 and 5. A low representation of the R··TC–Cbl complex is in line with the known values of its K_d_, ranging from approximately 0.2 to 4 nM ([1,7,35] and refs. thereof). These values are insufficient to saturate R, even at the maximal concentrations of TC-^57^Cbl present in the medium (100–220 pM), if assuming K_d_ = 1000 pM, and R ≈ 30 pM. A rough estimate gives the concentration of R··TC–^57^Cbl = 2.5–5 pM (equivalent to 1–2%), which cannot significantly affect the material balance of free TC-^57^Cbl and R (not bound to each other). Under these circumstances, the kinetics of Cbl uptake (A + B ↔ AB → C, where AB ≈ 0) resembles in its behavior a simple bimolecular reaction A + B → C.

We should point out that the internalized vitamin forms (CN[^57^Co]Cbl or HO[^57^Co]Cbl) did not differ in their retention inside lysosomes. Otherwise, this would have caused an elevated level of the unprocessed intracellular vitamin (where lysosomal Cbl is a part). Yet, the two ligands did not differ in this respect, because the “pooled” uptake constant k_3_ for CNCbl could be directly applied to the HOCbl model.

### 4.4. Kinetic Model: Steps 4 and 5, the Recirculation of the Unprocessed Cbl

If the freshly accumulated Cbl exceeds the intracellular capacity of its processing enzyme CblC (Figure 1), some part of Cbl is apparently excreted back to the medium without any transformation (steps 4 and 5 in Figure 5). A temporary stop point on the cell surface was added to explain a relatively high radioactivity, tightly associated with the membranes. A peculiar thing about this fraction concerns its stable level from 4 h to 48 h of incubation, despite a noticeable decrease of TC–Cbl in the medium. Such behavior can be explained by a recirculation route (steps 4 and 5). Equal parameters for CNCbl and HOCbl at step 5 (detachment from the cell surface) could not fit the experimental data, because the simulation curve for the surface HO[^57^Co]Cbl (Figure 2b, dashed line) was far below the experimentally measured surface radioactivity (Figure 2b, red circles). Therefore, we stipulated a slower release of HOCbl from the cell membranes, which also agreed with the results from refs. [19,20]. We ascribed this “hitch” of HOCbl on the cell surface to a well-known unspecific coordination of HOCbl to various protein groups, see, e.g., ref. [7,9] and references thereof. Step 5 (detachment) contains a “hidden” element—the interaction of Cbl with the surplus TC (a very fast process, limited by the rate of Cbl detachment from the cells). Finally, the newly produced TC–^57^Cbl enters another round of cellular uptake (step 3).

### 4.5. Kinetic Model: Step 6, the Reduction of the Internalized Cbl

This is a step of major interest. As was already mentioned, the retention of Cbl inside the cells is stipulated by the “competing” processes of its reduction (step 6) and the outward export (step 4). The adequate approximation of the experimental data in Figure 2a was observed at a six-fold faster reduction of HOCbl in comparison to CNCbl. It should be noted that this figure does not mean a six-fold better accumulation of HOCbl. In fact, the intracellular retention of Cbl is determined not by an individual rate constant but rather by the ratio of constants for step 6 (reduction) and step 4 (excretion). In the presented model (Table 2), the internalization of 24 molecules of CNCbl would end with the excretion of approximately 15 molecules and a reduction of only 9 molecules. In contrast, the processing of 23 molecules of the internalized HOCbl would bifurcate into five excreted and eighteen reduced molecules. Therefore, the overall excretion/retention ratio of CNCbl (18/23) is approximately two-fold higher than that of HOCbl (9/24).

### 4.6. Kinetic Model: Steps 7 and 8, the Synthesis of MeCbl and AdoCbl

We found that the production of MeCbl was considerably faster than the synthesis of AdoCbl, irrespective of the supplementation of CNCbl or HOCbl to the medium (Figure 4 and Table 2). The lower rate of AdoCbl synthesis was ascribed to a longer route, passed by the processed Cbl on its way from cytoplasm to the mitochondria, containing MMCM + CblB (Figure 5). In contrast to the “long” AdoCbl route, all conversions within the MeCbl branch take place in the cytoplasm. Therefore, the enzymatic pair of MS + CblE effectively intercepts CblD··Cbl, resulting in a 10-fold higher probability of MeCbl synthesis vs. AdoCbl (k_+7_ and k_+8_ in Table 2). We have earlier speculated [28] that a lower intracellular content of MS in comparison to MMCM [36] guarantees an early equilibration of MS with Cbl, whereupon the rest of the “excessive” Cbl is delivered to MMCM. Only at this point does the synthesis of AdoCbl start gaining momentum, while the production of MeCbl slows down. The current curves for MeCbl and AdoCbl synthesis agree with the aforementioned sequential mechanism for the appearance of Cbl coenzymes. The data from the literature show that CNCbl (when administered orally at physiological doses of <18 μg) was converted to MeCbl upon the passage from the human intestine to blood [37]. This observation strengthens our statement about the initial formation of MeCbl in cells.

The synthesis of MeCbl and AdoCbl was set as two reversible processes, where both coenzymes undergo a backward conversion to [Co^2+^]Cbl (constants k_−7_ and k_−8_ in Table 2). We made such a stipulation because the respective enzymes have rather low affinities for their Cbl ligands (Km = 340 nM for MS [38] and Kd = 80 nM for MMCM [39]), at least in comparison to the total intracellular Cbl, the concentration of which has the same order of magnitude. It should be mentioned, however, that k_−7_ and k_−8_ are not identical with the dissociation rate constants and include several reactions related to a general turnover of the cofactors. The values of backward parameters in steps 7 and 8 could not be assigned precisely because it was difficult to determine the end levels of MeCbl and AdoCbl in our current setup. Yet, our recent data for rat liver revealed the following Cbl composition: 35% MeCbl, 37% AdoCbl, 15% HOCbl, and 13% CNCbl for the animals on a standard diet (containing CNCbl) [28]. The chosen rate constants k_−7_ and k_−8_ (Table 2) provided a reasonably adequate composition of the Cbl forms in HeLa cells ending after 500 h with 40% MeCbl, 40% AdoCbl, and 20% [Co^2+^]Cbl (the latter purified as HOCbl).

### 4.7. Study Limitations

In our model, we assumed an irreversible accumulation of the processed Cbl by cells. Such an approximation (being not literally true) is sufficiently fair, because no leakage of HO/Me/AdoCbls to the medium was detected in the CNCbl setup (Section 3.4). In addition, previous human and rat studies revealed that the ratios of (endogenous tissue Cbl)/(plasma Cbl) ranged from approximately 10 to 100 ([1,40] and refs. thereof). Similar data were also observed during the accumulation of radioactive Cbls in rat liver traced over time. The ratio (tissue ^57^Cbl)/(plasma ^57^Cbl) corresponded to three or ten one day after the administration of CN[^57^Co]Cbl or HO[^57^Co]Cbl, respectively. These values increased to twenty-five (for CN[^57^Co]Cbl) or 220 (for HO[^57^Co]Cbl) after seven days ([40] and refs. thereof), thereby approaching an “irreversible” mechanism of Cbl uptake.

Other limitations of our study include the incomplete extraction of the total intracellular radioactivity (≈90%) and an exact classification of ≈80% of Cbls in the well-expressed HPLC peaks. In the absence of any better option, the fractions of classified Cbls were directly projected onto the total intracellular radioactivity. A few other minor shortcomings can be mentioned. Only one ^57^Cbl supplementation scheme was thoroughly tested (^57^Cbl = 0.22 nM, TC = 0.44 nM + 0.44 nM). The application of bovine TC to human cells (because of the ready availability of bTC and its good performance in an assay with Caco-2 cells [41]) might be also questioned. Yet, we should point out that the respective Cbl complexes of bovine and rabbit TC interacted with the receptors in the human placenta with equal affinities, resembling a similar experiment with human TC vs. rabbit TC ([1,7,35] and refs. thereof). In addition, it is quite acceptable that HeLa cells grow in the medium with 10% of the heated bovine serum, which is not exactly the conditions of human body.

### 4.8. Potential Inferences Relevant to B_12_ Treatment

We found that the uptake and cellular processing of CNCbl and HOCbl follow the same route with nearly identical rate coefficients except for (i) the reduction of Cbl in the cytoplasm (faster for HOCbl) and (ii) the detachment of Cbl from the cell surface upon its recirculation to the medium (slower for HOCbl). Different reduction rates for HOCbl and CNCbl are well known in chemical settings and reflect a noticeable difference in the electrochemical potentials of HOCbl (E_½_ = +0.2 V, indicating a higher tendency to reduction) and CNCbl (E_½_ = −0.76 V, indicating a more constrained reduction) [9,10]. This difference is seemingly applicable to the enzymatic processing of Cbls in vitro. Thus, the relevant information is presented in ref. [13], but the authors omit any explicit conclusion about the exact comparative kinetics, being engaged in the chemistry of the process. We examined the data of ref. [13], where the authors depict reduction of CNCbl and HOCbl by eCblC (CblC from *Caenorhabditis elegans*) and GSH under identical conditions (Figures 3c and 6a of ref. [13]). We found that the respective apparent rate constants of the two processes are k_obsCN_ ≈ 0.055 min^−1^ (deduced from GSH record in Figure 3c of ref. [13]) and k_obsHO_ = 1.27 min^−1^ (given in the main text of ref. [13], related to Figure 6a of ref. [13]). This comparison points out that at least one type of enzymatic reduction takes place considerably faster for HOCbl than for CNCbl. However, the simultaneous presence of several reductive routes inside the cell (involving CblC, flavins, MSR, NADPH, GSH, as well as free and protein-bound Cbl [13]) complicates any direct transition from in vitro to in vivo conditions.

In the current publication, we attempted to address this issue by applying a kinetic model to the intracellular processing of CNCbl vs. HOCbl (where all reductive routes act simultaneously) and found a faster reduction of HOCbl in vivo. This causes a better intracellular retention of HOCbl over 48 h. Yet, the overall uptake of both vitamins seems to be comparable, if extrapolating the curves to an infinite incubation time. Other works on the cellular uptake of Cbl are difficult to interpret in this regard because the absent surplus of TC in the medium hindered the recirculation of the effluxed Cbl. Therefore, we made a comparison to our earlier publications ([42] and refs. thereof) concerning the supplementation of vitamin pills with CNCbl vs. HOCbl (or HOCbl in pills/milk/milk fraction) to lactovegetarian Indians. These studies revealed a much longer extracellular life of CNCbl (2–10 weeks at higher doses of the vitamins), though the overall metabolic responses were concurrent in both groups (if starting monitoring one week after initiation of the treatment). The prolonged circulation of CNCbl was explained by its slower intracellular processing accompanied by the repeated excretion to blood, where CNCbl gradually accumulates on a slow-exchanging carrier called haptocorrin [42].

We speculate whether the inferences of our cell assay can be extended to clinical settings. For instance, a more rapid response (1–2 days) to the treatment of Cbl deficiency with HOCbl can be surmised. Yet, the final outcome (after a longer period of treatment, e.g., 7–10 days) is expected to be the same, irrespective of the vitamin form, at least for the normal genotypes. Our work provides an indication that the early response to the successful treatment might be a decrease in homocysteine (the substrate for MeCbl-dependent reaction), because MeCbl was the first cofactor produced by cells from both CNCbl and HOCbl. We should also point out that the increasing plasma B_12_ does not signify a success of CNCbl treatment, because CNCbl has a tendency to “linger” in circulation giving an overrated indication of the B_12_ uptake. This makes the analysis of total B_12_ an unreliable monitoring marker. The responses of metabolites (as well as changing clinical picture if present) are more robust indications of an improvement in the B_12_ status.

## 5. Conclusions

We show in the culture of HeLa cells that the rates of B_12_-uptake are not identical for the two typical forms of vitamin B_12_: CNCbl and HOCbl. A two-fold faster intracellular accumulation of radioactivity was observed during supplementation of radiolabeled HO[^57^Co]Cbl in comparison to CN[^57^Co]Cbl. Both vitamin-forms were gradually converted to the coenzymes MeCbl and AdoCbl (giving more coenzymes from HOCbl over 48 h). Kinetic modeling revealed that a good retention of HO[^57^Co]Cbl is associated with a high rate of its intracellular reduction, preceding formation of the coenzymes. If the absorbed vitamin is not reduced, it is excreted to the external medium. This recirculation was more expressed for CN[^57^Co]Cbl, because of its six-fold slower reductive processing in comparison to HO[^57^Co]Cbl. Therefore, a faster metabolic response can be surmised when treating B_12_ deficiency with a fast-accumulating HOCbl than with a slow-accumulating CNCbl (though both vitamins eventually tend to similar levels of the uptake). We also found that the intracellular MeCbl appeared much earlier than AdoCbl, irrespective of the supplementation of CNCbl or HOCbl. The logical consequence of the latter observation is that a decrease in homocysteine is the earliest indication of a successful treatment with B_12_.

## Figures and Tables

**Figure 1 nutrients-16-00378-f001:**
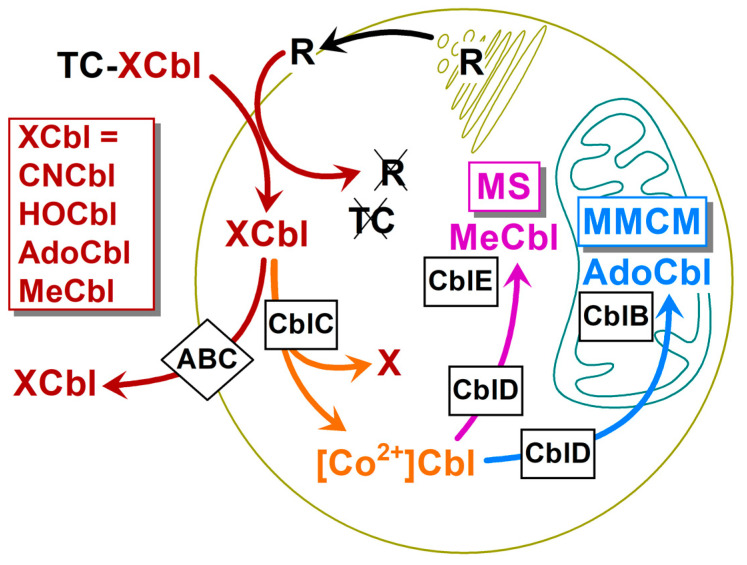
The intracellular processing of Cbl (notation of steps generally follows ref. [12]). Different Cbl forms (XCbl) circulate in blood, bound to an excess of the specific carrier transcobalamin (TC). The TC–XCbl complex enters the cell with the help of its specific receptor (R). Both protein moieties (TC and R) are endocytosed and degrade in lysosomes, whereupon free XCbl enters the cytoplasm with the help of a lysosomal membrane transporter (omitted in this figure). The liberated XCbl is reduced in the cytoplasm by the enzyme chaperon CblC and thus loses its original X-group. The superfluous XCbl leaves the cell unprocessed via the MRP1/ABCC1 transporter (ABC). The processed [Co^2+^]Cbl is delivered with the help of the adapter protein CblD to either cytoplasmic methionine synthase (MS) and MS reductase (CblE), or to the mitochondrial methylmalonyl-CoA mutase (MMCM) and adenosyltrasferase (CblB). The first pair of enzymes produces and utilizes MeCbl, while the second pair synthesizes and uses AdoCbl.

**Figure 2 nutrients-16-00378-f002:**
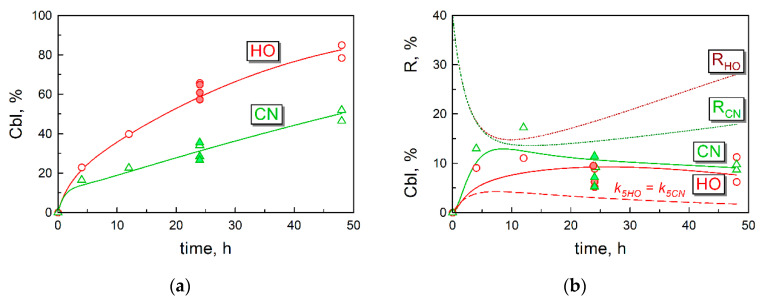
Time-dependent records for the cellular uptake of radioactive Cbls, normalized to 10^7^ cells (open symbols). Closed symbols notate the data of exploratory experiments (24 h), where the number of cells was assessed as 10^7^. (**a**) Intracellular Cbl. The experimental measurements are shown as the individual points for CN[^57^Co]Cbl (Δ, green) and HO[^57^Co]Cbl (○, red). The experimental dispersion was estimated as σ = 4.15%, see Appendix A. The solid lines (tagged as “CN” and “HO” for the respective Cbl-ligands) depict kinetic simulations according to the final kinetic model of Cbl uptake (see Section 3.5). Here, 100% of Cbl units % corresponds to 1.1 pmol of ^57^Cbl (bound to 2.2 pmol of TC) in 5 mL of the medium. (**b**) Tightly associated surface Cbl. The points correspond to measured surface radioactivity released after trypsinolysis of the cells. The experimental dispersion was estimated as σ = 2.44%. The solid lines show simulations based on the final kinetic model. The lower dashed curve (shown in red) depicts the turnovers of surface HO[^57^Co]Cbl if assuming k_5HO_ = k_5CN_, see Section 3.5 and Table 2. The turnover of the surface receptor R (dotted lines at the top) was theoretically simulated but not measured, see more in Section 4.3 of the Discussion. Abbreviations R_CN_ and R_OH_ reflect simulations of R-turnover in the experiments with CN[^57^Co]Cbl and HO[^57^Co]Cbl, respectively. The other notation is explained in the legend to panel (**a**).

**Figure 3 nutrients-16-00378-f003:**
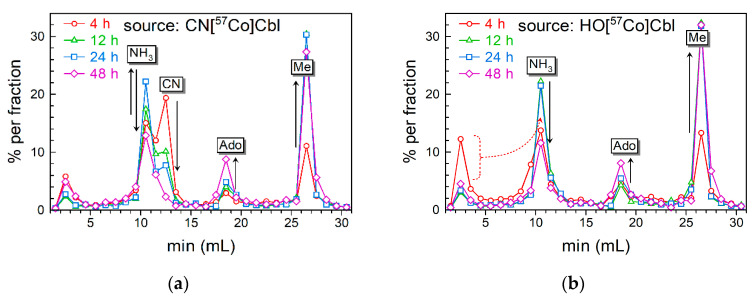
The HPLC elution profiles of the intracellular Cbls. Experiments with (**a**) CN[^57^Co]Cbl or (**b**) HO[^57^Co]Cbl. The records are presented as a percentage of radioactivity in each HPLC fraction, normalized to the total radioactivity in the profile. The time of incubation (cells + ^57^Cbl) is shown in the upper left corner of each panel. The tags above the peaks correspond to NH_3_Cbl (generated from HOCbl and its substituted variants), CNCbl, AdoCbl, and MeCbl. Solid arrows schematically depict transitions of the peaks during incubation of cells with ^57^Cbl (4–48 h). The dotted arrow in panel (**b**) indicates that a part of the breakthrough peak (1.5 min for 4 h of incubation) was assumed to be HOCbl, unspecifically bound to some peptides via cobalt coordination, see the main text for further details.

**Figure 4 nutrients-16-00378-f004:**
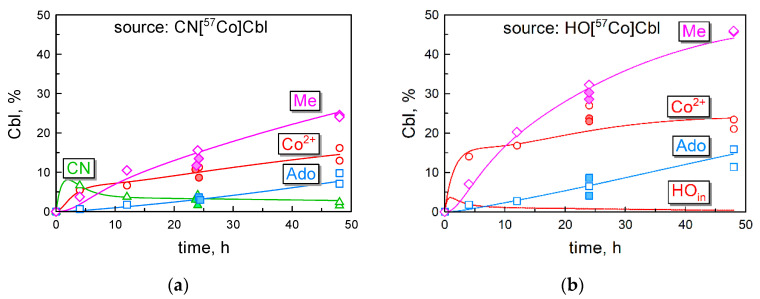
Intracellular conversions of the accumulated Cbls. Experiments with (**a**) CN[^57^Co]Cbl; and (**b**) HO[^57^Co]Cbl. Percent units % reflect fractions of each individual Cbl in the pool of total radioactivity added to the cells. Thus, 10% corresponds to ≈4.4 nM of the intracellular concentration (assuming 25 μL of intracellular volume for 10^7^ cells). Experimental measurements are shown as the individual points. The experimental dispersion of data was assessed as σ = 1.42% for panel a and σ = 2.15% for panel (**b**), see Appendix A. The curves depict our computer simulations based on the designed kinetic model (Figure 5). The tags “CN”, “Ado”, and “Me” correspond to the respective Cbl forms. The tag “Co^2+^” assembles all different reduction grades of Cbl in one category, isolated as HOCbl. The tag “HO_in_” in panel B represents the absorbed but unprocessed intracellular HO[^57^Co]Cbl, which cannot be experimentally differentiated from the pool of processed [Co^2+^]-forms. The representation of HO_in_ was assessed from the model simulations.

**Table 1 nutrients-16-00378-t001:** The distribution of radioactivity between different fractions after 24 h of incubation (HeLa cells + bTC−^57^Cbl complex).

Samples	CN[^57^Co]Cbl, %	HO[^57^Co]Cbl, %	*p* of Equal Values
Intracellular content	28.5 ± 3.6	57.6 ± 4.2	0.0008
Cell-free medium	47.4 ± 4.4	12.2 ± 0.7	0.002
Cell surface	7.9 ± 2.2	6.9 ± 1.5	0.98
Unaccounted loss	14.6 ± 2.1	19.8 ± 4.2	0.26

The data are shown as mean ± SEM (n = 3). The counts were not normalized to the number of cells.

## Data Availability

Data are contained within the article and Appendix A.

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
