# Peer review of "Kinetics of Cellular Cobalamin Uptake and Conversion: Comparison of Aquo/Hydroxocobalamin to Cyanocobalamin"

_nutrients, 2024, doi:10.3390/nu16030378_

Round 1

Reviewer 1 Report

Comments and Suggestions for Authors

This is an excellent piece of work conducted by experts in the field. The manuscript is very well written, accompanied by a detailed description and justification of each experiment carried out.

My expertise in cell work and computer modelling is limited, therefore I am not able to scrutinise all analytical and statistical approaches taken in this work (in particular those described in sections 2.3 and 2.6).   The editors may wish to obtain further feedback from someone more experienced in this area. However, even if my expertise does not exactly align with the type of work presented in this paper, I am happy with the level of detail, clear and transparent explanations of each step of the study and limitations presented.

Author Response

Thank you for your high evaluation of our work.

Reviewer 2 Report

Comments and Suggestions for Authors

In the article entitled: Kinetics of cellular cobalamin uptake and conversion: comparison of aquo/hydroxocobalamin to cyanocobalamin the authors take into consideration the biopharmaceutical and diagnostic problem i.e. the Vitamin B12 uptake. They used the in vitro model based on the HeLa cell line. As they correctly recognised the above scientific problem is important due to the homocysteine and methylmalonic acid metabolism. For their studies, they used the sensitive method used 57Co as a radioemitter. Moreover, they tried to extrapolate their data to physiological terms. From the current state of knowledge, the data is interesting and worth polishing. The references have been correctly cited and selected. However, I have several questions which should be answered and critical remarks which should be considered before publication.

-             In the abstract part, the medical meaning of the authors results must be introduced (the medical importance)

-             in the introduction, the nutritional uptake of B12 was omitted as well as the LADME of this vitamin, the ADI was also not mentioned. The nutritional level should be presented as age-dependent.

-             The metabolism of the cysteine and its toxicity should be added

-             The DNA damage as Vit B12 related must be mention

-             The answer to the question: why did authors decide to use the HeLa line instead normal one (not cancer)

-             The description of 57Co are highly required (gamma/beta emitter, t1/2, etc.)

-             The introduction part and results part are too long

-             The figure 1 and 5 are almost identical

-             The appendixes should be moved to supplementary materials

-             Please check carefully the text (typing mistakes)

Author Response

Answers to Reviewers

Reviewer 2

Comment 1. In the abstract part, the medical meaning of the authors results must be introduced (the medical importance).

Answer. We have slightly reformulated the two last sentences of the Abstract to address this request: “In medical terms, our data suggest: (i) an earlier response to the treatment of Cbl-deficiency with HOCbl; and (ii) manifestation of a successful treatment first as a decrease in homocysteine.”

Comment 2. In the introduction, the nutritional uptake of B12 was omitted as well as the LADME of this vitamin, the ADI was also not mentioned. The nutritional level should be presented as age-dependent.

Answer. We have added some basic data concerning the nutritional uptake of B12, both under normal conditions and during the treatment of deficiency. Yet, the extensive coverage of this subject (well-described in the cited literature) falls beyond the scope of our manuscript. We hope that Reviewer 1 approves the following text in the beginning of the revised Introduction: “Cobalamin (Cbl, B12) is an important nutrient required at a quantity of 2 – 3 µg per day [1–4]. … Ingestion of a normal physiological dose (2 – 10 µg) results in the maximal absorption of ≈ 1.5 – 2 µg via the specific transportation route [2,3,7,8]. The high oral doses of 1 – 5 mg are used for treatment of Cbl-deficiency and add to the specific uptake further 10 – 50 µg, which penetrate intestinal walls by diffusion [2–4,7].”

Comment 3. The metabolism of the cysteine and its toxicity should be added.

Answer. We add the requested information to the introduction and hope that the following text will suffice “Thus, a decreased activity of MS results in a low level of the major methylating precursor methionine, as well as in a high level of unmetabolized homocysteine [1–4]. The latter thiol is frequently associated with inflammation, oxidative stress and microvascular diseases [3]. In addition, trapping of folate in its “inactive” methyltetrahydrofolate form hampers DNA synthesis [3,4]. Low activity of the second Cbl-dependent enzyme MMCM leads to accumulation of methylmalonic acid and a general acidification of the body [3,4].”

Comment 4. The DNA damage as Vit B12 related must be mention.

Answer. We address this subject in connection to our answer to comment 3.

Comment 5. Why did authors decide to use the HeLa line instead normal one (not cancer).

Answer. We are aware that the study presents some limitations related to the use of a single cell line, the HeLa cells. However, the human cervical carcinoma cell line represents a well-established and widely used model in the studies on vitamin B12 metabolism. Further studies in other cellular / animal models are of course advisable, but the model used here allows for comparison with the previous works on B12 metabolism published using the HeLa model.

No changes to the text have been introduced.

Comment 6. The description of 57Co are highly required (gamma/beta emitter, t1/2, etc.).

Answer. We have added a short statement to Section 2.3 “The isotope 57Co emits γ-radiation (t½ = 272 days), which was counted in the scintillation cocktail OptiPhase HiSafe 3 using the energy window of 0 – 200 keV.

Comment 7. The introduction part and results part are too long.

Answer. We made an attempt to shorten the text, though these attempts were compromised by addition of a new text to answer the comments of two Reviewers, who pointed out a few unclear subjects.

Comment 8. The figure 1 and 5 are almost identical.

Answer. We understand this point but see no other option, because Fig. 1 presents the enzymes involved in turnovers of Cbl, while Fig. 5 sketches out our kinetic scheme and its steps. Presentation of Figures 1 and 5 as two panels in the same figure is not possible, because the data in Fig. 1 are generally known from the literature (making it more suitable for the Introduction), while Fig. 5 presents the final kinetic model (with a logical place at the end of manuscript).

No changes have been made.

Comment 9. The appendixes should be moved to supplementary materials.

Answer. The advised change has been done.

Comment 10. Please check carefully the text (typing mistakes).

Answer. The revised text has been inspected, and several typing and formatting errors have been corrected.

Reviewer 3 Report

Comments and Suggestions for Authors

The present manuscript from the Fedosov laboratory deals with the uptake and distribution of cobalamins by mammalian cells.  Although these types of studies have been reported extensively in the past, there are some novel developments, which merit the publication of the study.  The authors used HeLa cells incubated with 57Co-enriched cobalamin, either in the form of CN-Cbl or HO-Cbl.  They were able to monitor the levels of the radioactive vitamin in the extra-cellular media, adsorb onto the cell surface, and internalized into the HeLa cells, by re-working the protocol for isolation and extraction of the vitamins, resulting in a >80% recovery.  In particular, the ability to recover intact HO-Cbl through avoiding a high temperature evaporation step is noteworthy.  The methods are described in enough detail, and the models are well rationalized. Furthermore, the data were analyzed by means of a simplified steady state model that accounts for most of the intermediates involved in transcobalamin (TC) complex formation, cell uptake, intracellular release of Cbl, alkylation of Cbl by CblD-MS or MMCM.  The model and kinetic equations appear to be correct and are as simple as required by the provided dataset.

I do have, however, some concerns, which I describe below:

1. The experiments were set up to mimic the natural uptake of CN-Cbl with supplementation with bovine trans-cobalamin. However, the utilized cells (HeLa) are human cancer cells, which are acceptable models for non-cancerous mammalian cells.  Would it not be more appropriate to use human trans-cobalamin (TC) for these experiments?  Does the human homolog degrade under the media conditions?

2. Why was there a need to add more bTC after 24 hours?  Is apo-bTC susceptible to proteolysis or inhibition after 24 hours?

3. Is NH4OAc required for the extraction of cobalamins in this protocol?  Assuming a HOAc/NaOAc buffer is being used, would that prevent the formation of the fast eluting NH3-Cbl complex, from HO-Cbl?  I can see the utility of converting minor forms of Cbl into a single peak, but I prefer to have a protocol that does not affect the distribution of species.

4. How does the ligand, 5-amino-tetrazole, promote the extraction of HO-Cbl without affecting the other cobalamins? Is ATZ lost during the chromatography and is there an explanation for that?

5. Line 28 should read “nutrient synthesized by”, or something close, line 131 should read Sigma.  In line 222 and subsequent cases, the “u(%)” is not correct, ratios or percentages have no units.  The symbol % should be sufficient.

6.  In line 223, if Cout(t=0) is 0, is the experiment corrected for the addition of TC-Cbl complexes? Or are the rates of TC-X-Cbl complex formation slow enough to matter during the time-scale of the experiments?

7. Can the authors explain why the turnover of the R receptor, which was not monitored, was altered by the nature of the X ligand?  I would think that the recovery rates would be independent of the ligand present since the de novo synthesis of the receptor is not regulated by cobalamins.

8. Related to 7, is the inclusion of turnover of R necessary for the model to fit the data to their model?

9. Are the y-axis scales in both Figures, 2a and 2b normalized as mol 57Co per mol of cells?  If not then what is 100% saturation?  The maximal concentration that is indicated is about 0.22 nM, but the caption states that is normalized on a per 10^7 cells.

10.  The fast eluting peak at 1.5 min was observed both with HO-Cbl and CN-Cbl, but only the former decays at latter time, indicating that the nature of X matters in its reactivity.  That would make sense but it is contrary to the explanation provided in lines 344-354.  It seems that some of the cobalamins are reacting with components in the intracellular compartments of the cells, CN-Cbl for example, but only the HO-Cbl can be subsequently converted to the free form or other alkylated forms.

11.  Related to the same issue, if the pool at 1.5 min is saturated, then that should have been revealed by a concentration dependence on the concentration of TC-X-Cbl.

12. Related to 11, in lines 416-417 the authors explained that the values for k7 and k8 (as well as k-7 and k-8) should have no effect on the uptake of Tc-X-Cbl or the membrane concentrations.  But a better experimental approach would have been to look at the TC-X-Cbl dependence of the apparent rate constants, which if the model is correct, would predict no effect of those concentrations on the synthesis of Ado-Cbl and Me-Cbl.

13. In line 456 replace “undisclosed” with “unresolved”.  In line 463 the word “frame” should be replaced with “framework”

14. I do not follow the 486-488 calculations.  You start with a 220 pM concentration of 57Co-Cbl ligand, but 40% comes from the actual surface counts?  Or is 40% the result of subtracting the recovered radioactivity minus intracellular concentration? The closest I can come up with is %R = 100% - (unrecovered counts)% - (Cell free medium)% which is about 38%.

15. The study assume Cout (t=0)= 100%.  That makes sense if the concentration of TC-X-Cbl complexes are much higher than the dissociation constants.  But Cin might be more than 0%, since HeLa cells start with some cobalamin in them.  Is there a way to estimate the amount of non-radioactive cobalamins at t=0, by HPLC and do they matter?

16.  Had the authors compared the radioactivity recovered as HOCbl at the different evaporation temperatures?

17. In line 549, the authors refer to an earlier publication where the MS enzyme concentration is much lower than that of MMCM enzyme, so MeCbl equilibrates much faster relative to AdoCbl.  But if that is the case, shouldn't MeCbl then decrease at later times as more AdoCbl is synthesized by MMCM+CblE, or is the biosynthesis of either cofactor irreversible?

18. Did the authors consider monitoring the off-rates for intracellular Cbl pools being released into the dilute media (k5CN and k5OH), in the presence or absence of TC after 24 hours?  If the uptake is reversible, that would probably indicate that HeLa cells coupled with bTC is not a good model for in vivo scenarios.

19. The fastest steps in the proposed mechanism  (Table 2) appear to be the reduction of X-Cob(III)alamins to Cob(II) alamins.  The subsequent conversions to MeCbl or AdoCbl appear to be much slower but reversible.  I would think that the rates of return of Cob(II)alamin to the cell surface would be disfavored, this because TC can only bind the oxidized cob(III)alamin forms.  Is this correct?

Comments on the Quality of English Language

It is good enough.

Author Response

Answers to Reviewers

Reviewer 3.

Comment 1. The experiments were set up to mimic the natural uptake of CN-Cbl with supplementation with bovine trans-cobalamin. However, the utilized cells (HeLa) are human cancer cells, which are acceptable models for non-cancerous mammalian cells. Would it not be more appropriate to use human trans-cobalamin (TC) for these experiments? Does the human homolog degrade under the media conditions?

Answer. Bovine transcobalamin (TC) was ready available at the beginning of our study, while application of human TC required its expression and purification. Our previous work about the transcellular passage of Cbl through a monolayer of Caco-2 cells (see ref. 50) demonstrated an excellent performance of bovine TC, which reduces possible concerns about application of this protein. In addition, this is quite acceptable that HeLa cells grow in the medium with 10 % of heated bovine serum, which is not exactly the conditions of human body. Yet, we fully admitted in our manuscript, that application of bovine TC is a limitation (Section 4.7).

We have added one more comment at the very and of Section 4.7.

Comment 2. Why was there a need to add more bTC after 24 hours? Is apo-bTC susceptible to proteolysis or inhibition after 24 hours?

Answer. The presented model assumes recirculation of some portion of the internalized but unprocessed Cbl into the external medium. This would cause a repeated binding of Cbl to the surplus apoTC and cellular absorption of the newly formed TC-Cbl complex. After a number of cycles, the originally added apoTC will be exhausted. Therefore, a fresh portion of apoTC was added after 24 h. We did not add a considerable excess of apoTC, however; because apoTC binds to cd320 receptor, though with a lower affinity in comparison to TC-Cbl (holoTC). We also discuss, that application of only one Cbl-feeding scheme is a limitation. Yet, a thorough analysis of another feeding scheme would have caused a considerable increase in the size of our manuscript.

We have added a short clarifying sentence to Section 3.2 “More unsaturated bTC was added to the medium at 24 h to compensate its loss, after the unprocessed Cbl had been excreted to the medium and repeatedly delivered to the cells with help of bTC (see Section 3.5).”

Comment 3. Is NH4OAc required for the extraction of cobalamins in this protocol? Assuming a HOAc/NaOAc buffer is being used, would that prevent the formation of the fast eluting NH3-Cbl complex, from HO-Cbl? I can see the utility of converting minor forms of Cbl into a single peak, but I prefer to have a protocol that does not affect the distribution of species.

Answer. NH3Ac/HAc buffer was used because all its components are volatile. Substitution of NH3Ac/HAc buffer by its NaAc/HAc counterpart would give NaOH after evaporation, because HAc and (Ac + H+) largely leave the solution, while Na+ and OH are concentrated. A strong alkaline medium is not favorable for preservation of Cbl. Formation of NH3Cbl coordination complex from both HO/H2OCbl and the pool of its substituted forms (containing exchangeable ligands like GSH, His, ATZ, etc) requires a mildly alkaline pH (e.g. 9.3) of (NH3+H2O↔NH4OH)/HAc buffer, where the concentration of uncharged NH3 molecules is relatively high (pK 9.3). The produced NH3Cbl complex is reasonably firm and dissociates extremely slowly even under unfavorable conditions (e.g. acidic pH used during HPLC). The fraction of NH3Cbl on HPLC-profiles was regarded as a “pooled” HOCbl in our work. We see no problem in such an approach, because the true intracellular composition of the so calledHOCbl pool” (counting here also the oxidized [Co2+]Cbl and [Co1+]Cbl, as well as GSCbl, HisCbl, etc) is not known. For instance, the original [Co1+, 2+]Cbls (which are oxidized to H2O[Co3+]Cbl) can easily end as GSCbl, peptide-His-Cbl, peptide-S-Cbl, etc, if the respective compounds are present in the extract. In such way, any oxidating extraction misrepresents the true intracellular balance of Cbls. We simply pool all Cbls with potentially exchangeable groups because their internal balance is uncertain. In contrast to the aforementioned exchangeable upper ligands, the “strong” ligands (like those in MeCbl, AdoCbl, CNCbl, and SO3Cbl) remain unaffected during the treatment with ammonium, as was tested in model mixtures.

We have slightly reformulated the original Section 2.3 (its last paragraph, second half) to stress the difference between “reversible” and strongly bound Co-coordinating groups. The text now runs as follows: “… solids were dissolved in 0.15 – 0.25 mL of 0.2 M NH4Acetate buffer, pH 9.3, and heated for 10 min at 95 °C to convert potentially present HOCbl, ATZCbl and GSCbl (with reversibly bound Co-ligands) to a more stable NH3Cbl [10]. Formation of the latter complex improved separation of the original HO/ATZCbl forms from CNCbl on HPLC, see section 2.5. An individual test with pure Cbls showed that the ammonium treatment converted most of GSCbl and all HOCbl to NH3Cbl, but this procedure did not affect fractions of CNCbl, AdoCbl, MeCbl and SO3Cbl, which contain high-affinity ligands bound to the Co-site.”

Comment 4. How does the ligand, 5-amino-tetrazole, promote the extraction of HO-Cbl without affecting the other cobalamins? Is ATZ lost during the chromatography and is there an explanation for that?

Answer. In short, ATZ has a reasonably high affinity for the Co-site of Cbl (when the latter is in its aquo-form at pH 4 – 8), and the on-off rate constants of ATZ are high [35]. In other words, this ligand can, on one hand, displace Cbl from a number of reversible complexes (e.g. protein-His··[Co3+]Cbl), but on another hand, it promptly dissociates during exchange with any ligand with a higher affinity / higher concentration (e.g. uncharged NH3).

We have added a few modifications to the original text of Section 2.3: “The organic phase (0.5 mL) was mixed with equal volumes of chloroform and acetone, followed by addition of 0.4 mL 0.32 M NH4Acetate buffer, pH 4.6, containing 1 mM 5-aminotetrazole (ATZ). The sample was vigorously shaken and incubated for 20 – 24 h at 37 °C under rotation and complete protection from light. Presence of a “reversible” ligand ATZ with a reasonably high affinity for the Co-site [35] decreased by > 75 % association of HOCbl with the middle layer of denatured proteins during backward transition of Cbl from organic to aqueous phase in a model experiment. The effect was caused by a gradual dissociation of unspecific protein-Cbl coordination complexes, e.g. protein-His··[Co3+]Cbl [10,36], and formation of soluble ATZ··Cbl.” The substitution of ATZ by NH3 at the Co-site of Cbl was covered in our answer to Comment 3.

Comment 5. Line 28 should read “nutrient synthesized by”, or something close, line 131 should read Sigma. In line 222 and subsequent cases, the “u(%)” is not correct, ratios or percentages have no units. The symbol % should be sufficient.

Answer. We are sorry for this mix-up. The original line 28 contained an accidental deletion, which occurred during formatting of the text to the template of Nutrients. We have corrected this and other typos, and thank Reviewer 2 for his/her vigilance.

Commen 6. In line 223, if Cout(t=0) is 0, is the experiment corrected for the addition of TC-Cbl complexes? Or are the rates of TC-X-Cbl complex formation slow enough to matter during the time-scale of the experiments?

Answer. We think that this is a misunderstanding. The original line 223 runs as follows “… [Cout] = 100 u(%), the extracellular XCbl bound to TC …“, at least on our computers. In other words, the added radioactivity is equal to 100 % and not to 0 %. Is it possible that Reviewer 2 got a damaged pdf-file?

Comment 7. Can the authors explain why the turnover of the R receptor, which was not monitored, was altered by the nature of the X ligand? I would think that the recovery rates would be independent of the ligand present since the de novo synthesis of the receptor is not regulated by cobalamins.

Excellent comment! Indeed, this issue deserves a short explanation, and we have added the following text to Section 4.3: “It should be noted that the curves of R are not identical for CNCbl and HOCbl in the time range of 10 – 50 h (Figure 2b). This difference is caused by a higher concentration of TC-CNCbl in the medium, maintained at an elevated level because the intracellular processing of CNCbl is not as efficient as for HOCbl (step 6 in Figure 5), and a considerable part of the accumulating CNCbl is excreted (step 4 in Figure 5). The recirculated CNCbl binds to TC, whereupon TC-CNCbl (Cout) drives Rout into the cell according to the law of mass action for the reaction Cout + Rout Cin + Rin.

Comment 8. Related to 7, is the inclusion of turnover of R necessary for the model to fit the data to their model?

Answer. Yes, the turnover of R is necessary to explain the initial jump of the uptake for both CNCbl and HOCbl (Figure 2a, 0 – 5 h), followed by deceleration of the uptake at a later period of time (caused by exhaustion of R on the surface, shown in Figure 2b, two upper curves).

Comment 9. Are the y-axis scales in both Figures, 2a and 2b normalized as mol 57Co per mol of cells? If not then what is 100% saturation? The maximal concentration that is indicated is about 0.22 nM, but the caption states that is normalized on a per 10^7 cells.

Answer. We have modified the original text in Section 3.2 to explain this issue. “Therefore, we … normalized both the intracellular and the surface-bound radioactivity to a “standard” number of cells (n = 1 × 107, found at 24 h) to compare the samples collected at different time points. This means that the radioactive counts of the intracellular and surface fractions were divided by factors 0.8, 0.9, and 1.0 for the values of n = 0.8 × 107, 0.9 × 107, and 1.0 × 107, respectively. The counts are given in percent, where 100 % corresponds to the total radioactivity added to the medium at the start.

We hope that such an explanation will suffice.

Comment 10. The fast eluting peak at 1.5 min was observed both with HO-Cbl and CN-Cbl, but only the former decays at latter time, indicating that the nature of X matters in its reactivity. That would make sense but it is contrary to the explanation provided in lines 344-354. It seems that some of the cobalamins are reacting with components in the intracellular compartments of the cells, CN-Cbl for example, but only the HO-Cbl can be subsequently converted to the free form or other alkylated forms.

Answer. We think that this is a misunderstanding. Indeed, a small portion (3 – 5 %) of the extracted Cbl (both for CNCbl and HOCbl supplement) is always eluted in the break-through peak. The effect is most likely caused by some eluting capacity of the extract, which removes 3 – 5 % of any Cbl from the column. Percentage of this break-though Cbl does not change systematically upon increase of CNCbl inside the cells. In contrast, HOCbl shows a break-through peak of 12 % (of the absorbed quantity) at the lowest amount of HOCbl loaded into the cells. In contrast, this abnormally high peak decreases to its “usual” size of 3 – 5 %, when the amount of intracellular HOCbl increases. It seems that the extract has some additional binding capacity for HOCbl (but not for CNCbl), and this capacity is visible only at a very low content of the intracellular HOCbl. In other words, the additional binding compound exists as a minor contamination. What could be its nature? HOCbl differs from CNCbl by its ability to coordinate to various nucleophilic groups. We largely prevent such interactions by converting HOCbl (as well as a number of other substituted forms of HOCbl) to a firm complex of NH3Cbl. Yet, we cannot exclude that some compounds (e.g. peptides with thiol- and/or sulfate-groups) resist NH3 treatment and retain Cbl. Such coordination is impossible for CNCbl, because cyanide cannot be displaced from the Co-site of Cbl in course of any ordinary competition between two or more ligands.

We have added a few additional explanations to the last paragraph of Section 3.3.

Comment 11. Related to the same issue, if the pool at 1.5 min is saturated, then that should have been revealed by a concentration dependence on the concentration of TC-X-Cbl.

We assume that this unknown component of the break-through fraction (e.g. “peptide-SO3” with a high affinity for HOCbl) reacts with HOCbl nearly irreversibly. In other words, if the content of HOCbl is less than “peptide-SO3”, all HOCbl will be eluted at 1.5 min. If HOCbl exceeds the quantity of “peptide-SO3”, some part of HOCbl (matching “peptide-SO3”) will be eluted at 1.5 min, while the rest will be eluted as free NH3Cbl. If HOCbl considerably exceeds “peptide-SO3”, the fraction of peptide-bound HOCbl becomes negligible in comparison to NH3Cbl. The detailed investigation of this phenomenon (quite interesting but rather irrelevant) is clearly beyond the scope of our study.

Comment 12. Related to 11, in lines 416-417 the authors explained that the values for k7 and k8 (as well as k-7 and k-8) should have no effect on the uptake of Tc-X-Cbl or the membrane concentrations. But a better experimental approach would have been to look at the TC-X-Cbl dependence of the apparent rate constants, which if the model is correct, would predict no effect of those concentrations on the synthesis of Ado-Cbl and Me-Cbl.

Answer. Let us consider the scheme A → (B → C ↔ D), where the pooled content of B + C + D is measured (alike the absorbed intracellular Cbl). Such scheme is reduced to A → Z, where Z = B + C + D. The rate constants within the braced pool of B, C, D are of no importance for the accumulation of Z, because only the first rate constant of the full scheme governs conversion of A to Z. On the other hand, the “hidden” constants of B, C, D fraction are important for the time curves and the equilibrium levels of solely B, C, D, but not for those of Z. We think that this example is sufficiently illustrative.

No change to the text has been introduced.

Comment 13. In line 456 replace “undisclosed” with “unresolved”. In line 463 the word “frame” should be replaced with “framework”.

Answer. Corrections are done.

Comment 14. I do not follow the 486-488 calculations. You start with a 220 pM concentration of 57Co-Cbl ligand, but 40% comes from the actual surface counts? Or is 40% the result of subtracting the recovered radioactivity minus intracellular concentration? The closest I can come up with is %R = 100% - (unrecovered counts)% - (Cell free medium)% which is about 38%.

Answer. We have modified the text in question to “This value can be interpreted as the apparent concentration of R = 88 pM (see Section 3.5 concerning the connection between % and the real concentrations or quantities).” The original text of Section 3.5 (plus a small additional comment) explains this connection as follows “The model uses relative units, where 100 % corresponds to 0.22 nM concentration of 57Cbl in the external medium of 5 mL (i.e. quantity of 1.1 pmol), bound to bTC = 0.44 nM (replenished by adding more bTC = 0.44 nM at 24 h). All units for other interacting compounds are related to the concentration of 57Cbl in the medium, and e.g. R = 20 % corresponds to R-concentration of 0.044 nM.

Comment 15. The study assume Cout (t=0)= 100%. That makes sense if the concentration of TC-X-Cbl complexes are much higher than the dissociation constants. But Cin might be more than 0%, since HeLa cells start with some cobalamin in them. Is there a way to estimate the amount of non-radioactive cobalamins at t=0, by HPLC and do they matter?

Answer. We are sorry for this confusion, because our definition of Cout, Cin, and Csurf was not precise and can be misinterpreted. We have changed the text to a more accurate form, stressing that these are the radioactive forms of Cbl: “The concentrations of reactants (all in percent units) at the starting point (incubation time 0 h) were as follows: [Cout] = 100 %, the extracellular radioactive cobalamin X[57Co3+]Cbl bound to TC (i.e. TC-X[57Co3+]Cbl complex); [Cin] = 0 %, the unprocessed radioactive X[57Co3+]Cbl inside the cells; [C2] = 0 %, the processed radioactive [57Co2+]Cbl inside the cells; [Csurf] = 0 %, the unprocessed radioactive X[57Co3+]Cbl recirculated from the intracellular space to the cell surface; …” Obviously enough, the cells lack the radioactive 57Cbl at the start. Of course, the cells have some amount of endogenous Cbl, necessary for their development, and they were grown in the medium with “… 10% heat-inactivated fetal bovine serum (Sigma-Aldrich), final B12 ≈ 15 pM, free binding capacity ≈ 20 pM …” (as is mentioned in Section 2.2). Yet, this quantity is quite low in comparison to the added radioactive 57Cbl = 220 pM. Therefore, we do not assume any significant interference (competition) from the non-radioactive Cbl, if talking about the real biochemical system of our assay. The computer model does not assume such a competition, because all reaction follow the fist or second order without complex-formation.

Comment 16. Had the authors compared the radioactivity recovered as HOCbl at the different evaporation temperatures?

Answer. We have not tested different temperatures during evaporation, but in our recent work (ref. 31) we tested the effect of temperature, pH and presence of thiols on preservation of different Cbl-forms during their extraction from rat liver and in model mixtures. The results for rat liver were reproduced in Martin Warren’s lab (m.j.warren@kent.ac.uk) according to his private communication. As we do not think that introduction of vacuum would give any impact (except of concentration the original solution), the results of ref. 31 can be projected to the evaporation procedures without any significant corrections.

Comment 17. In line 549, the authors refer to an earlier publication where the MS enzyme concentration is much lower than that of MMCM enzyme, so MeCbl equilibrates much faster relative to AdoCbl. But if that is the case, shouldn't MeCbl then decrease at later times as more AdoCbl is synthesized by MMCM+CblE, or is the biosynthesis of either cofactor irreversible?

Answer. Yes, decrease of MeCbl is expected after 50 h of incubation and the designed model takes this into account. The highest level of (MeCbl = 47 % at 50 h) is transient and the content of this cofactor decreases to 40 % if modeling is stretched to 500 h. The original text at the very end of Section 4.6 makes a clear statement about this effect “… our recent data for rat liver revealed the following Cbl-composition: 35 % MeCbl, 37 % AdoCbl, 15 % HOCbl and 13 % CNCbl for the animals on a standard diet (containing CNCbl) [31]. The chosen rate constants k−7 and k−8 (Table 2) provided a reasonably adequate composition of the Cbl-forms in HeLa cells ending after 500 h with 40 % MeCbl, 40 % AdoCbl and 20 % [Co2+]Cbl (the latter purified as HOCbl).

Comment 18. Did the authors consider monitoring the off-rates for intracellular Cbl pools being released into the dilute media (k5CN and k5OH), in the presence or absence of TC after 24 hours? If the uptake is reversible, that would probably indicate that HeLa cells coupled with bTC is not a good model for in vivo scenarios.

Answer. Such effect was examined in ref. 20 on human fibroblasts (Fig. 5, bottom panels). The loss of CN57Cbl and HO57Cbl (loaded in advance with help of human plasma) stopped after 2 h at the respective levels of 19 % and 5 % in control cells. No further liberation of radioactivity was recorded over 24 h. In contrast, CblC mutants released after 2 h CN57Cbl 50 % and HO57Cbl 15 %, and this release continued further (though at a lower rate). This experiment indicates that a critical step of the “irreversible” retention is situated at the CblC-reaction. At that point, the loading protein TC is already degraded and cannot give any effect. Reversibility of Cbl loading carried out by bovine TC vs human TC is very dubious, considering the curves of Cbl adsorption (Fig. 2a of the submitted manuscript). Both curves tend to a level of above 90 %. In addition, we tested “leakage” of the processed Cbls from cells at 24 h of incubation with CN57Cbl by analyzing Cbl-forms in the medium. Only CN57Cbl was found, with no trace of HOCbl, MeCbl or AdoCbl.

A comment has been added at the end of Section 3.4.

Comment 19. The fastest steps in the proposed mechanism (Table 2) appear to be the reduction of X-Cob(III)alamins to Cob(II) alamins. The subsequent conversions to MeCbl or AdoCbl appear to be much slower but reversible. I would think that the rates of return of Cob(II)alamin to the cell surface would be disfavored, this because TC can only bind the oxidized cob(III)alamin forms. Is this correct?

Answer. The reduction of HOCbl is, indeed, the fastest step in the whole turnover, while reduction of CNCbl is much closer to other constants. The scheme in Fig. 5 does not consider export of Cob(II)alamin to the surface, which generally agrees with ref. 20. Our test of Cbl-forms in the medium 24 h after beginning the CNCbl experiment revealed presence of only CNCbl. This indicates that the processed Cbls (HOCbl, MeCbl and AdoCbl, generated after CblC-reaction) do not leak from the cells, at least within the time interval of 1 – 2 days. We assume that TC can bind [Co2+]Cbl, perhaps with a lower affinity. The reduced form has no “upper” group X (which is seemingly of no consequence for the binding). It also has a decreased affinity for the “lower” nucleotide base, resembling the adenosyl-form of pseudo B12 (Ado-pB) at a neutral pH (doi: 10.1021/bi062063l). Yet, TC binds Ado-pB with an apparent equilibrium dissociation constant of 1.7 × 10-13 (doi: 10.1021/bi062063l). Such affinity is quite enough to bind a ligand at picomolar concentrations

.

Round 2

Reviewer 2 Report

Comments and Suggestions for Authors

The provided correction I recommend the article for publication. The answers to my question are sufficient.